# The Microwave Absorption in Composites with Finemet Alloy Particles and Carbon Nanotubes

**DOI:** 10.3390/ma15228201

**Published:** 2022-11-18

**Authors:** Anatoly B. Rinkevich, Dmitry V. Perov, Evgeny A. Kuznetsov, Yulia V. Korkh, Anna S. Klepikova, Yury I. Ryabkov

**Affiliations:** 1M.N. Miheev Institute of Metal Physics UB RAS, Sofia Kovalevskaya St., 18, 620108 Ekaterinburg, Russia; 2Institute of Chemistry UB RAS, Pervomayskaya St., 48, 167000 Syktyvkar, Russia

**Keywords:** microwave properties, nanocomposites, transmission and reflection coefficients, ferromagnetic resonance and antiresonance, Finemet alloy

## Abstract

The absorption of waves of the centimeter and millimeter wavebands in composites with Finemet alloy particles and carbon nanotubes has been studied. It has been established that ferromagnetic resonance and antiresonance are observed in such composites. A method is proposed for calculating the effective dynamic magnetic permeability of a composite containing both a random distribution of ferromagnetic particles and a part of the particles oriented in the same way. In the approximation of effective parameters, the dependences of the transmission and reflection coefficients of microwaves are calculated. It is shown that the theoretical calculation confirms the existence of resonant features of these dependences caused by ferromagnetic resonance and antiresonance. The theory based on the introduction of effective parameters satisfactorily describes the course of the field dependence of the coefficients and the presence of resonance features in these dependences. The frequency dependence of the complex permittivity of the composite is determined. The dependence of the complex magnetic permeability on the magnetic field for millimeter-wave frequencies is calculated.

## 1. Introduction

Composites with conductive particles are a popular object of study and find important practical applications. The preparation technologies and the information about the electrical and magnetic properties of composites with conductive particles were described in the monograph [1]. The study of the magnetic permeability is fundamentally important for calculating the microwave absorption. The mechanisms of microwave absorption in granular structures with ferromagnetic nanoparticles in a dielectric matrix were studied in [2]. It has been established that magnetic-type losses arise due to the rapid relaxation of the spin of nanoparticles. Finemet FeCuNbSiB alloy was often used as a material for conducting ferromagnetic particles. The measurement of a composite containing flake-shaped FeCuNbSiB particles in a paraffin matrix in the frequency range from 0.1 to 18 GHz was carried out in [3]. It was shown that composites with a preferred orientation of the flakes has a higher magnetic permeability. Using the Landau-Lifshitz-Gilbert equation in conjunction with Bruggeman’s method, the effective magnetic permeability was calculated [3]. The dynamic magnetic permeability of the Finemet alloy tape in weak magnetic fields was measured at frequencies of 100 kHz–1.8 GHz [4]. The results of studying the magnetic properties of Finemet-type alloys with different chemical compositions were given in [5].

Microwave absorption, magnetic permeability and dielectric permittivity of a composite containing Finemet type alloy flakes in an epoxyamine matrix were studied in [6]. The microwave magnetic permeability in this composite under the application of an external magnetic field was studied in [7]. The permeability calculation takes into account the concentration of particles, their shape and orientation, including the presence of a predominant orientation in some part of the particles. It was shown that the phenomena of ferromagnetic resonance (FMR) and antiresonance (FMAR) were observed in the composite. The dependences of the power dissipation and the wave penetration depth on the external magnetic field were determined. The microwave refractive index was studied in the same composite in [8]. It was found that in the absence of an external magnetic field and in weak fields, the composite behaves like a lossy dielectric, which is completely predictable for a composite up to the percolation threshold. However, in fields near the FMR, the imaginary part of the refractive index becomes of the same order as the real part, i.e. the composite becomes similar to a conducting medium.

A new method for studying the microwave magnetic properties of films and composites was proposed in [9]. The sample was placed in a coaxial line, a constant magnetic field was applied, and the generator excited an electromagnetic field with a swept frequency. As a result of measurements, magnetization values and information about magnetic anisotropy can be obtained. The FMR in amorphous FeSiBNbCu ribbons obtained by fast melt quenching was studied in [10]. It has been established that the value of magnetization decreases monotonically with an increase in the concentration of boron and copper. The effective magnetic permeability of a material containing a core–shell-like structure, where the core is a ferromagnetic particle and the shell is a conductive overlayer, was studied theoretically in [11]. The conductive sheath partially shielded the magnetic particle, because of which the permeability changes.

Significant efforts have been made to obtain composite materials with good radio absorption and to investigate their properties [12,13]. The microwave absorbing characteristics of the FeNiMo/epoxy composite with spherical particles were studied at frequencies of 2–18 GHz in [14]. As the ratio of the particle axes increases, both the real and imaginary parts of the magnetic permeability and dielectric permittivity increased. It was shown in [15] that cobalt particles in the form of nanosheets 120 nm thick have excellent radio absorbing properties in the X band, i.e., at frequencies from 8 to 12 GHz. It is known that the addition of a carbon component to the composite enhances the shielding properties. Multi-walled carbon nanotubes [16,17,18], graphite [19], and carbon black [20] were added, always with a positive result. Most of the above works dealt with the X-band frequencies. An absorber with a polarization dependence was proposed in [21] using a graphene-polymer film.

This review [22] presents the designing of a single wire functional unit and assembling it in a specific arrangement to develop a range of microwire composites made of Co_60_Fe_15_Si_10_B_15_ alloy with predictable electromagnetic responses. The composites with vertical interface on randomly dispersed short-cut microwires are developed in [23]. This design allows the adjustment of electromagnetic properties to a large extent.

In this paper, we considered the electromagnetic properties of composite materials containing Finemet-type alloy flakes and carbon nanotubes (CNT) in an epoxyamine matrix. Nanotubes were added to the composite in order to increase dielectric losses. The frequency range from 26 to 38 GHz was chosen for the study. This range was chosen so that FMR and FMAR phenomena could be observed in one experiment. The FMAR phenomena is not observed at lower frequencies. In our experiment, the method of transmission and reflection of microwaves, interacting with a composite plate, was realized. The study aimed to reveal similarities and differences in microwave properties in the composites with CNT compared to the composites without CNT. Therefore, a comparison was made with the results of [7,8], where similar composites were studied without CNT. The study of the structure of composites, the phase composition of Finemet flakes and the measurement of magnetic properties were carried out. In the approximation of effective parameters, the dependences of the transmission and reflection coefficients of microwaves are calculated. It is shown that the theoretical calculation confirms the existence of resonant features of these dependences caused by ferromagnetic resonance and antiresonance. The theory based on the introduction of effective parameters satisfactorily describes the course of the field dependence of the coefficients and the presence of resonance features in these dependences. The frequency dependence of the complex permittivity of the composite is determined. The dependence of the complex magnetic permeability on the magnetic field for millimeter-wave frequencies is calculated.

## 2. Sample Preparation and Characterization

The particles in the form of flakes of the Fe-Si-Nb-Cu-B alloy were used to prepare the composite. The chemical analysis of the particles was performed with the Horiba 500 X-ray fluorescence spectrometer, as well as by the atomic emission spectroscopy with inductively coupled plasma. The elemental composition of the flakes is as follows: about 77% Fe, 8.5% Si, 8.4% Nb, 1.1% Cu, 1.2% B, 0.2% Cr, 0.13% Mn, 0.06% Ni. It is a Finemet-type alloy with high magnetic permeability. The distribution of average particle radii is shown in Figure 1a. The average particle radius, that is, the average distance from the center of mass of a particle to its boundary, is 20.1 µm. The maximum Martin diameter, that is, the maximum length of the median line passing through the grain, is on average 51 µm, and the minimum Martin diameter is on average 27 µm. The ratio of these diameters is 1.9; this characterizes the mean ratio of lateral sizes.

The composite material was prepared by mechanical mixing of the particles in an epoxy resin matrix Poly(bisphenol-A-co-epichlorohydrin) Liquid Epoxy resin (Biphend A type). Epoxy resin was chosen due to its moderate permittivity. After mechanical stirring, processing was carried out in an ultrasonic bath. After that, a liquid mixture of the epoxy resin and particles was poured into metal mandrels. In these mandrels the size of the cavity exactly equals the transverse dimensions of the waveguides in which microwave measurements will be carried out. For example, for the frequency range from 26 to 38 GHz, the cavity size is 7.112 mm × 3.556 mm. The curing and polymerization processes of the mixture continued for several hours. The specimens were hold at 80 °C for 1 h, then at 120 °C for 1 h and, finally, at 160 °C for 3 h. Heating is realized at the rate of 5 °C per minute. Two series of composite samples with 15 wt. % flakes and 3.8 wt. % flakes + 2% CNT were prepared. The X-ray phase analysis was performed with a spectrometer from Pananalytical with the vertical high-resolution goniometer for the phase analysis with high productivity and accuracy and measurement of the lattice parameters. The diffractometer was equipped with X’Pert HighScore Plus v.3.0 software. The analysis showed that the main phases are 2 bcc phases of the α-Fe type (f1 и f2), differing only in lattice parameters, which are equal to 2.871 Å and 2.841 Å. The X-ray diffraction pattern for a sample with 15% particles is shown on Figure 1b. The splitting of the peaks in Figure 1b can be caused by a slight tetragonal distortion of the crystal lattice of bcc iron.

The structure of the composite was studied using the Quanta 200 scanning electron microscope (Columbus, OH, USA) from FEI with the EDAX instrument for the elemental analysis and 5 nm resolution at an accelerating voltage of 30 kV. Figure 2 shows the structure of the composite with 3.8% Fe-Si-Nb-Cu-B flakes + 2% CNT, and the images were taken from the cleavage, that is, from the inside of the sample (Figure 2a), as well as from the upper surface of the sample (Figure 2b). It can be noted that in the inner part of the composite the particles are randomly oriented, while near the upper surface there is a preferential orientation of the flakes parallel to the surface. The preferred orientation probably arose due to the action of surface tension during the solidification process. We also note that there is no electrical contact between the particles, so the direct current conductivity is extremely low. The elemental composition in the region of the particle, marked with a red square in Figure 2b, was obtained by the method of energy dispersive analysis (see Table 1).

Figure 3 shows the magnetization curve of a composite material containing 3.8% flakes and 2% CNT. The measurements were performed with a VSM 7407 vibrating magnetometer. In fields above $\mu_{0}H_{s}\approx 0.5$ T, magnetic saturation was observed, which was to be expected for ferromagnetic particles. To check the anisotropy of the magnetic properties of the sample, which could arise during the preparation of the composite, we measured the magnetization curves of the cubic sample in three mutually perpendicular directions. The results turned out to be almost identical, so there is no magnetic anisotropy of the composite.

## 3. Microwave Transmission and Reflection Method

Microwave measurements were performed at the frequencies from 12 to 38 GHz according to the technique described in [7,8]. The calibration of the network analyzer is carried by using the SOLT calibration technique [24]. The scheme of the experiment is shown in Figure 4. The sample was placed in the cross section of a standard rectangular waveguide in such a way as to completely overlap the waveguide cross section. The thickness of the samples *d* was from 1 to 2 mm. The waveguide operates on the TE_10_ mode, and the operating frequency range is determined by its dimensions. For example, a WR-28 waveguide with a cross section of 7.112 mm × 3.556 mm was used for the interval 26–38 GHz, and a WR-15 waveguide with a cross section of 3.759 mm × 1.880 mm was used for the interval 53–78 GHz. The incident wave propagates along the normal to the sample’s surface. The measurements were carried out using the scalar network analyzer. The modules of the transmission coefficient *T* and the reflection coefficient *R* and their frequency dependences were measured. Since the phases of the waves were not measured, then by the coefficients *T* and *R* we mean their moduli. Measurements of the coefficients were used to determine the complex permittivity $\dot{\varepsilon }=\varepsilon^{\prime }-i\varepsilon^{\prime \prime }$.

Let us briefly describe the procedure for determining the complex permittivity using the scalar network analyzer. The complex transmission and reflection coefficients were calculated by the formulas [8,25]:(1)$$\dot{T}=\frac{2Z_{1}{\dot{Z}}_{2}}{2Z_{1}{\dot{Z}}_{2}\cos\;\left({\dot{k}}_{2}d_{2}\right)+i\;\left[Z_{1}^{2}+{\dot{Z}}_{2}^{2}\right]\;\sin{\left({\dot{k}}_{2}d_{2}\right)}^{}{}^{}}$$
(2)$$\dot{R}=\frac{i\;{\left[{\dot{Z}}_{2}^{2}-Z_{1}^{2}\right]}_{}^{}\sin\left({\dot{k}}_{2}d_{2}\right)}{2Z_{1}{\dot{Z}}_{2}\cos\;\left({\dot{k}}_{2}d_{2}\right)+i\;\left[Z_{1}^{2}+{\dot{Z}}_{2}^{2}\right]\sin{\left({\dot{k}}_{2}d_{2}\right)}^{{}^{}}}$$

In Formulas (1) and (2), medium “1” is the inner space of the waveguide. Medium “2” is a sample, namely, an imperfect ferromagnetic dielectric with thickness $d_{2}$. The waveguide complex impedance ${\dot{Z}}_{2}$ for it can be written as follows:(3)$${\dot{Z}}_{2}=\frac{\omega \mu_{0}\dot{\mu }}{{\dot{k}}_{2}}$$
where $\mu_{0}$ is the permeability of vacuum, ω=2π f is the circular frequency.

The waveguide impedance for medium “1” is
(4)$$Z_{1}=\sqrt{\frac{\mu_{0}}{\varepsilon_{0}}}\frac{1}{\sqrt{1-{\left(\frac{\pi \;c}{\omega \;a}\right)}^{2}}}$$
where *ε_0_* is the permittivity of vacuum, *c* is the speed of light in vacuum, *a* is the larger transverse dimension of the rectangular waveguide, Formula (3) include complex parameters of the sample: magnetic $\dot{\mu }=\mu^{\prime }-i\mu^{\prime \prime }$ permeability and dielectric $\dot{\varepsilon }=\varepsilon^{\prime }-i\varepsilon^{\prime \prime }$ permittivity. The complex wave number for an imperfect ferromagnetic dielectric in (1) and (2) is calculated by the formulas [26].
(5)$${\dot{k}}_{2}={k^{\prime }}_{2}-i{k^{\prime \prime }}_{2}$$
$${k^{\prime }}_{2}=\sqrt{\frac{\sqrt{{ℜ}^{4}+{ℑ}^{4}}+{ℜ}^{2}}{2}}$$
$${k^{\prime \prime }}_{2}=\sqrt{\frac{\sqrt{{ℜ}^{4}+{ℑ}^{4}}-{ℜ}^{2}}{2}}$$
$$ℜ=\sqrt{{\left(\frac{\omega }{c}\right)}^{2}\left(\varepsilon^{\prime }\mu^{\prime }-\varepsilon^{\prime \prime }\mu^{\prime \prime }\right)-{\left(\frac{\pi }{a}\right)}^{2}}$$
$$ℑ=\frac{\omega }{c}\sqrt{\left(\varepsilon^{\prime \prime }\mu^{\prime }+\varepsilon^{\prime }\mu^{\prime \prime }\right)}$$

The power transmission *T_p_* and reflection *R_p_* coefficients are introduced as follows:$$T_{p}=\dot{T}\cdot T^{*}$$
(6)$$=\frac{4Z_{1}^{2}Z_{2}^{*}{}^{2}}{{\left[2Z_{1}{\dot{Z}}_{2}\cos\;\left({\dot{k}}_{2}d_{2}\right)+i\;\left[Z_{1}^{2}+{\dot{Z}}_{2}^{2}\right]\sin\left({\dot{k}}_{2}d_{2}\right)\right]}^{}\left[2Z_{1}Z_{2}^{*}\cos\;\left(k_{2}^{*}d_{2}\right)-i\;\left[Z_{1}^{2}+Z_{2}^{*}{}^{2}\right]\sin\left(k_{2}^{*}d_{2}\right)\right]}$$
$$R_{p}=\dot{R}\cdot R^{*}$$
(7)$$=\frac{{\left[{\dot{Z}}_{2}^{2}-Z_{1}^{2}\right]}_{}^{}\left[Z_{2}^{*}{}^{2}-Z_{1}^{2}\right]\sin\left({\dot{k}}_{2}d_{2}\right)\;\sin\left(k_{2}^{*}d_{2}\right)}{{\left[2Z_{1}{\dot{Z}}_{2}\cos\;\left({\dot{k}}_{2}d_{2}\right)+i\;\left[Z_{1}^{2}+{\dot{Z}}_{2}^{2}\right]\sin\left({\dot{k}}_{2}d_{2}\right)\right]}^{}\left[2Z_{1}Z_{2}^{*}\cos\;\left(k_{2}^{*}d_{2}\right)-i\;\left[Z_{1}^{2}+Z_{2}^{*}{}^{2}\right]\sin\left(k_{2}^{*}d_{2}\right)\right]}$$
where the asterisk is the sign of complex conjugation. Knowing the coefficients (6) and (7), we can determine the dissipation coefficient, which shows the fraction of the microwave power dissipated in the sample:(8)$$D=1-T_{p}-R_{p}$$

The dissipation of microwave power is carried out due to absorption in the sample, scattering on internal inhomogeneities of the sample material, and transformation into evanescent modes on the sample, as on inhomogeneity.

We assume that the magnetic permeability is known and consider the procedure for finding the complex permittivity from the measured frequency dependences of the moduli of the transmission and reflection coefficients. Let us denote the experimentally measured frequency dependence of the modulus of the transmission coefficient as ${\tilde{T}}_{p}(\omega \dot{,}\dot{\varepsilon },\dot{\mu })$ and the modulus of the reflection coefficient as ${\tilde{R}}_{p}(\omega ,\dot{\varepsilon },\dot{\mu })$. Let us write down the difference between the theoretically calculated $T_{p}$ and the measured one ${\tilde{T}}_{p}$ as $\Delta_{T}={\tilde{T}}_{p}(\omega ,\dot{\varepsilon },\dot{\mu })-\;T_{p}(\omega ,\dot{\varepsilon },\dot{\mu })$ and similarly for the reflection coefficient the difference $\Delta_{R}={\tilde{R}}_{p}(\omega ,\dot{\varepsilon },\dot{\mu })-R_{p}(\omega ,\dot{\varepsilon },\dot{\mu })$. The complex permittivity $\dot{\varepsilon }$ is an unknown quantity. To find it, the choice of $\dot{\varepsilon }$ is made to achieve a minimum of the value
(9)$$\Delta =\underset{\dot{\varepsilon }}{\min}\;\left\{\int_{\omega_{\min}}^{\omega_{\max}}\left[\sqrt{{\left(\Delta_{R}\left(\omega ,\dot{\varepsilon },\dot{\mu }\right)\right)}^{2}+{\left(\Delta_{T}\left(\omega ,\dot{\varepsilon },\dot{\mu }\right)\right)}^{2}}\right]\;d\omega \right\}$$
within the selected angular frequency range from $\omega_{\min}$ to $\omega_{\max}$, upon condition $\dot{\mu }=1$, in the sense of the least squares method. The resulting value $\dot{\varepsilon }$ is considered an estimate of the permittivity. To perform the minimization procedure, a frequency interval is selected in which the amplitude-frequency characteristics of transmission and reflection coefficients are known. For this interval, the operating frequency interval of the waveguide can initially be selected. If at any frequency within a given interval the difference between $T_{p}$ and ${\tilde{T}}_{p}$, as well as $R_{p}$ and ${\tilde{R}}_{p}$ does not exceed a predetermined value (i.e., the error in determining $\dot{\varepsilon }$), then it is considered that the obtained value $\dot{\varepsilon }$ is constant within the error. This happens if the frequency dispersion of $\dot{\varepsilon }$ is small. If this difference exceeds the specified error, then the determination of $\dot{\varepsilon }$ and the minimization procedure are repeated using a sliding frequency window. The minimization procedure (9) is performed at each position of the sliding window, and as a result, the frequency dependence of $\dot{\varepsilon }(\omega )$ is obtained. From the obtained values of
$\dot{\varepsilon }$, the microwave conductivity $\sigma =\varepsilon_{0}\varepsilon^{\prime \prime }\omega$ can be found.

## 4. Transmission and Reflection Coefficients

The frequency dependences of the moduli of transmission and reflection coefficients were measured for composite samples with 3.8% Finemet particles and 2% CNT in the following frequency intervals: from 26 to 38 GHz and from 53 to 78 GHz, in each interval using an appropriate waveguide. It turned out that it is required to use the processing technique with a sliding frequency window, and as a result, the dependence $\dot{\varepsilon }(\omega )$ was obtained. Figure 5 shows the frequency dependences of the coefficients *R* and *T*, measured (thin curves) and calculated (thick curves) for a composite with 3.8% Finemet particles and 2% CNT in the frequency ranges of 26–28 GHz and 53–78 GHz. The thickness of this sample is *d* = 1.7 mm. Depending on the frequency dispersion $\dot{\varepsilon }(\omega )$, the width of the sliding window was chosen from 0.4 to 4 GHz. In this case, the difference between the approximating and measured dependences of *T* and *R* did not exceed 0.02. The results of calculation of the permittivity are presented below.

The frequency dependences of the real and imaginary parts of the permittivity of the composite with 3.8% flakes and 2% CNT in the frequency range from 26 to 38 GHz are shown in Figure 6a, and in the frequency range from 53 to 78 GHz are shown in Figure 6b. The presence of a maximum near 29 GHz in the frequency dependence of the imaginary part indicates an increased dissipation of the microwave power near this frequency. A summary of estimates of the complex permittivity results and specific electrical conductivity of the composite with 15% flakes, averaged over the measurement frequency ranges, is given in Table 2. The values of the dielectric constant of the epoxyamine matrix are also given there. It can be seen from these data that the addition of Finemet particles significantly increases both the real and imaginary parts of the permittivity. The microwave conductivity of the composites is low, approximately 2 S/m.

The microwave magnetic permeability of a composite with 15% flakes at H = 0 was studied in [6]. It was found there that in the frequency range under consideration, the real part of the magnetic permeability does not exceed 1.04, and the imaginary part is less than 0.1. Of course, for the composite with 3.8% flakes and 2% CNT, the real part of the magnetic permeability was even closer to 1.0, and the imaginary part was also less than 0.1. Significant changes in the magnetic permeability in a magnetic field near FMR conditions will be considered below in the Discussion section.

## 5. Results of Microwave Measurements in a Magnetic Field

Microwave measurements in a magnetic field were carried out with the orientation of magnetic fields $H⊥H_{\sim }$**_,_** where H and $H_{\sim }$ are the vectors of DC and microwave magnetic fields, respectively. With such an arrangement of vectors, FMR can be observed. At several frequencies from the range of 26–38 GHz, the field dependences of the transmission and reflection coefficients were measured. The results for the composite with 3.8% flakes and 2% CNT are shown in Figure 7. From now we shall use the symbols on the curves only for their identification. The dependences of the transmission coefficient show a minimum caused by wave absorption under FMR conditions. Minima are also observed in the dependences of the reflection coefficient (see Figure 7b). In addition to the minima, the dependences also contain the maxima, which occur in the fields smaller than the FMR field. These maxima are caused by FMAR [7]. The FMAR phenomenon has been firstly observed in ferromagnetic metal films and is caused by the vanishing of the real part of the effective magnetic permeability [26]. In composite materials, this phenomenon has significant features [7,27]. For metal films, the skin depth sharply increases during FMAR, and therefore the transmission coefficient increases. In contrast, in nanocomposites prepared on the basis of a dielectric matrix and not exhibiting the skin effect, a change in the impedance during FMAR leads to a maximum reflection coefficient. It may be important to fulfill the conditions of a quarter-wave plate. The maximum in the reflection coefficient is then especially strong [28]. It is also known that the FMAR phenomenon cannot be observed at frequencies below that characteristic of a given material depending on the saturation magnetization.

Note that the positions of the minima in the dependences of the transmission and reflection coefficients measured at the same frequency are different. For example, at a frequency *f* = 35 GHz, the transmission coefficient has a minimum at *H* = 10 kOe. The minimum of the reflection coefficient at this frequency is not reached even at the maximum field of 12 kOe. The true FMR field in micro-inhomogeneous media should be determined from the dissipation maximum *D* (8). The field dependence of dissipation for a composite with 3.8% flakes and 2% CNT is shown in Figure 8. This figure shows both experimental data on dependences with empty symbols and theoretically calculated dependences with filled symbols. Note that the largest dissipation in Figure 8 corresponds to frequencies of 38 GHz (this is the highest frequency) and 29 GHz, which accounts for the maximum of the imaginary part of the permittivity in Figure 6a. Thus, the results obtained from the frequency dependences of the coefficients at *H* = 0 and the results obtained in a magnetic field confirm each other.

## 6. Discussion

Earlier, in Section 2, Formulas (1), (2), (6) and (7) were given, from which the field and frequency dependences of the transmission and reflection coefficients can be calculated. The calculation requires data on the dielectric and magnetic permeability, which are included in Formula (3). The permittivity data are obtained experimentally and are shown in Table 2. In this section, we first calculate the effective magnetic permeability. A method for calculating the magnetic permeability of a composite, which takes into account the tensor nature of the permeability and is suitable for the case of an external magnetic field, was developed in [7]. Let us briefly describe the calculation algorithm as applied to a composite with non-spherical magnetic particles, some of which are randomly oriented in space, and there are also groups of particles with the same orientation.

The calculation of the magnetic permeability values is performed for an ensemble of ferromagnetic particles of the “flakes” type, consisting of 2000 particles. Of these, 1600 particles are randomly oriented and another 400 are oriented in the plane of the sample. For different frequencies, the values of the diagonal ${\dot{\mu }}_{xx}$ and off-diagonal ${\dot{\mu }}_{xy}$, ${\dot{\mu }}_{yx}$ components of the complex magnetic permeability tensor are determined, as well as the complex effective magnetic permeability ${\dot{\mu }}_{eff}$, which is calculated by the formula:(10)$${\dot{\mu }}_{eff}={\dot{\mu }}_{xx}-\frac{{\dot{\mu }}_{xy}{\dot{\mu }}_{yx}}{{\dot{\mu }}_{yy}}$$

For a composite medium containing differently oriented ferromagnetic particles, Formula (10) is rewritten as follows:(11)$$\langle {\dot{\mu }}_{eff}\rangle =\langle {\dot{\mu }}_{eff}\left(\Theta \right)\rangle =\langle {\dot{\mu }}_{xx}^{m}\left(\Theta \right)-\frac{{\dot{\mu }}_{xy}^{m}\left(\Theta \right)\cdot {\dot{\mu }}_{yx}^{m}\left(\Theta \right)}{{\dot{\mu }}_{yy}^{m}\left(\Theta \right)}\rangle$$

The angle brackets mean averaging over the ensemble. Let the particles have the form of flakes. The spatial orientation of a particle can be characterized by the direction of the normal to its flat surface. For example, if the normal is along the *y*-axis, then the normal vector is $n={\left(\begin{array}{ccc} 0 & 1 & 0 \end{array}\right)}^{T}$. We believe that this is the basic orientation of the ferromagnetic flake, for which the vector of rotation angles relative to the coordinate axes can be defined as $\Theta_{0}=\left(\begin{array}{ccc} 0 & 0 & 0 \end{array}\right)$. The spatial positions of the particles will then be given by the vectors $\Theta =\left(\begin{array}{ccc} \alpha & \beta & \gamma \end{array}\right)$. If the particles are randomly oriented, each of the elements of the set of vectors **Θ** belongs to the sets of uniformly distributed random numbers belonging to the intervals: α∈[−π ; π], β∈[−π ; π] and γ∈[−π ; π]. Discrete sets of triples of numbers $\alpha_{p}$, $\beta_{p}$, $\gamma_{p}$. correspond to a discrete set of values of the rotation angle vector $\Theta_{p}$. Taking into account the presence of randomly oriented particles and selected groups of particles, Formula (11) can be rewritten as
$$\langle {\dot{\mu }}_{eff}\rangle =\frac{L_{2}}{L_{1}+L_{2}}\left({\dot{\mu }}_{xx}^{m}\left(\Theta_{0}\right)-\frac{{\dot{\mu }}_{xy}^{m}\left(\Theta_{0}\right)\cdot {\dot{\mu }}_{yx}^{m}\left(\Theta_{0}\right)}{{\dot{\mu }}_{yy}^{m}\left(\Theta_{0}\right)}\right)$$
(12)$$+\frac{1}{L_{1}+L_{2}}\sum_{p=1}^{L_{1}}\left({\dot{\mu }}_{xx}^{m}\left(\Theta_{p}\right)-\frac{{\dot{\mu }}_{xy}^{m}\left(\Theta_{p}\right)\cdot {\dot{\mu }}_{yx}^{m}\left(\Theta_{p}\right)}{{\dot{\mu }}_{yy}^{m}\left(\Theta_{p}\right)}\right)$$

In (12), the first term in the numerator describes the contribution of randomly oriented particles, and the second term describes the contribution of groups of particles. The number of particles in oriented groups is $L_{1}$, and that of randomly oriented groups is $L_{2}$. The total number of particles is $L_{1}+L_{2}$. In our case, $L_{1}$ = 400, $L_{2}$ = 1600. Formulas for calculating the components of the magnetic permeability tensor are given in [7]. They are written for a magnetized medium at $H>H_{s}$ and magnetization M‖H



(13)
$$\mu_{xx}^{m}\left(\Theta \right)=1+\frac{\theta_{v}\;\omega_{M}\left(\Theta \right){\left[\omega_{H}+i\omega \alpha -\left({\tilde{N}}_{zz}\left(\Theta \right)-{\tilde{N}}_{yy}\left(\Theta \right)\right)\;\left(1-\theta_{v}\right)\;\omega_{M}\left(\Theta \right)\right]}^{}}{\hat{D}\left(\Theta \right)}$$


$$\mu_{xy}^{m}\left(\Theta \right)=\frac{\theta_{v}\omega_{M}\left(\Theta \right){\left[i\omega -{\tilde{N}}_{12}\left(\Theta \right)\left(1-\theta_{v}\right)\;\omega_{M}\left(\Theta \right)\right]}^{}}{\hat{D}\left(\Theta \right)}$$


$$\mu_{yx}^{m}\left(\Theta \right)=-\frac{\theta_{v}\omega_{M}\left(\Theta \right){\left[i\omega +{\tilde{N}}_{12}\left(\Theta \right)\left(1-\theta_{v}\right)\;\omega_{M}\left(\Theta \right)\right]}^{}}{\hat{D}\left(\Theta \right)}$$


$$\mu_{yy}^{m}\left(\Theta \right)=1+\frac{\theta_{v}\;\omega_{M}\left(\Theta \right){\left[\omega_{H}+i\omega \alpha -\left({\tilde{N}}_{zz}\left(\Theta \right)-{\tilde{N}}_{xx}\left(\Theta \right)\right)\;\left(1-\theta_{v}\right)\;\omega_{M}\left(\Theta \right)\right]}^{}}{\hat{D}\left(\Theta \right)}$$


$$\hat{D}\left(\Theta \right)={\left[\omega_{H}+i\omega \alpha -\left({\tilde{N}}_{33}\left(\Theta \right)-{\tilde{N}}_{11}\left(\Theta \right)\right)\;\left(1-\theta_{v}\right)\;\omega_{M}\left(\Theta \right)\right]}^{}$$


$$\cdot \left[\omega_{H}+i\omega \alpha -\left({\tilde{N}}_{33}\left(\Theta \right)-{\tilde{N}}_{22}\left(\Theta \right)\right)\;\left(1-\theta_{v}\right)\;\omega_{M}\left(\Theta \right)\right]-{\left({\tilde{N}}_{12}\left(\Theta \right)\left(1-\theta_{v}\right)\;\omega_{M}\left(\Theta \right)\right)}^{2}-\omega^{2}$$



In formula (13) $\omega_{H}=\gamma \mu_{0}H_{z}$ $\omega_{M}\left(\Theta \right)=\gamma \mu_{0}M_{z}\left(\Theta \right)$; $\overset{↔}{N}$ is the tensor of the demagnetizing factors, $\overset{↔}{N}=\frac{1}{4\pi }\cdot \overset{↔}{N}$, $\mathrm{tr}\overset{↔}{N}=1$. In our calculation, the volume fraction of the ferromagnetic phase is *θv* = 0.038. The saturation magnetization of Finemet is $M_{s}$ = 0.9 MA/m, and $M_{z}=M_{s}$ was assumed in the calculations because $M_{x}<<M_{z}$ and $M_{y}<<M_{z}$ [7]. The dissipation parameter in the magnetic system α was determined for each frequency based on the condition that the widths of the theoretical and experimental lines of the frequency dependences of the absorption coefficients coincide. A typical value in calculations is α = 0.2. Calculations using Formulas (12) and (13) give the results for the complex effective magnetic permeability of the composite shown in Figure 9. The changes in the value of the real part of the magnetic permeability are small, and the course of the field dependences has a typical character. The imaginary part of the permeability has a maximum, the field of which shifts to stronger fields with increasing frequency.

Figure 10 compares the measured and theoretically calculated field dependences of the transmission and reflection coefficients for a nanocomposite with 3.8% flakes and 2% CNT. The comparison was made for frequencies of 26 and 38 GHz. The choice of these two frequencies is due to the following reasons. For the frequency of 26 GHz, there is no clearly pronounced maximum on the dependence of the reflection coefficient, while such a maximum exists for 38 GHz. The course of dependences and the presence of maxima and minima are similar for the calculated and measured dependences. Thus, a qualitative agreement between the calculation and experiment has been achieved. The existing quantitative differences between the calculated and measured dependences are probably caused by two circumstances. The first is that the Finemet flakes used have a wide size distribution, which is not taken into account in the calculation. A broad size distribution can, in principle, lead to some inhomogeneity in the volume distribution of the ferromagnetic phase. The second circumstance is related to the use of Formulas (1) and (2), which for a strongly inhomogeneous medium with effective parameters should be considered as approximate ones.

The microwave measurements prove that the composites, which contain magnetic flakes and carbon nanotubes, can be used in the microwave devices of the millimeter waveband, where the microwave properties tuned by the magnetic field are required.

## 7. Conclusions

The passage of microwaves through a plate made of a composite material containing Finemet alloy flakes and carbon nanotubes and reflections from it have been experimentally and theoretically studied. The frequency and field dependences of the transmission and reflection coefficients are measured. The complex permittivity of the composite has been estimated in the frequency ranges from 12 to 38 and from 53 to 78 GHz. The microwave measurements have been carried out in magnetic fields up to 1.2 T.

A method is proposed for calculating the complex magnetic permeability of a composite material containing groups of ferromagnetic particles with a random spatial orientation and groups with a certain orientation of particles. The dependence of the complex magnetic permeability on the magnetic field for millimeter-wave frequencies is calculated.

In the approximation of effective parameters, the dependences of the transmission and reflection coefficients of microwaves on the magnetic field are calculated. It is shown that the theoretical calculation confirms the existence of resonant features of these dependences caused by ferromagnetic resonance and antiresonance. One of the main results on the paper is that the relatively simple theory, based on the introduction of effective parameters, describes satisfactorily the character of the field dependences of the transmission and reflection coefficients and their resonance features for the complex composite media, containing magnetic metallic flakes and carbon nanotubes.

The composites, which contain magnetic flakes and carbon nanotubes, can be used in the microwave devices of the millimeter waveband. The method for calculating the complex magnetic permeability and the transmission and reflection coefficients of a composite material can be applied to get the composite media with predetermined microwave properties.

## Figures and Tables

**Figure 1 materials-15-08201-f001:**
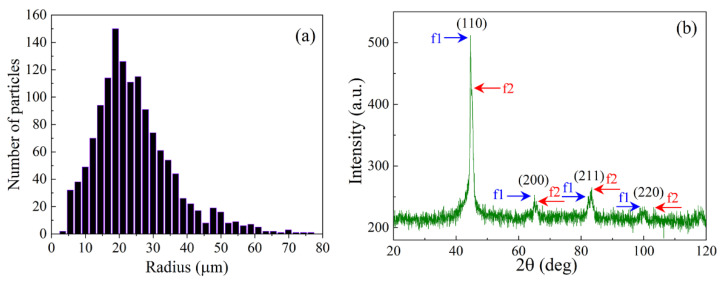
The distribution of flake radii (**a**); the X-ray diffraction image of the composite medium for a sample with 15% particles (**b**).

**Figure 2 materials-15-08201-f002:**
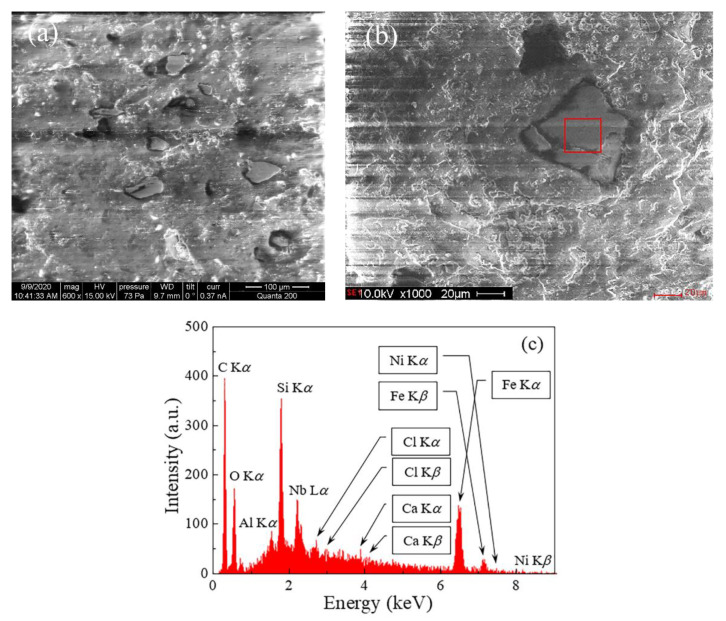
Structure of the composite medium with 3.8% Fe-Si-Nb-Cu-B flakes + 2% CNT (**a**); the section of the composite on which the energy dispersive analysis was performed (**b**); the result of energy dispersive analysis (**c**).

**Figure 3 materials-15-08201-f003:**
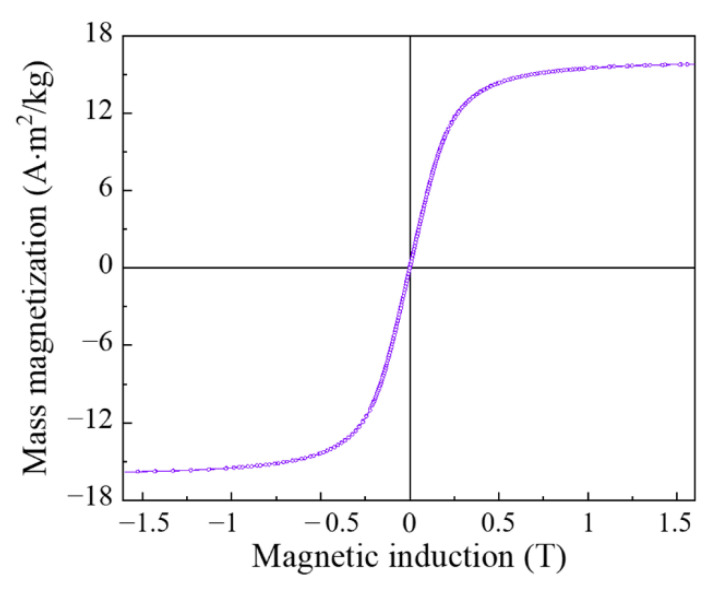
Magnetization curve of a composite material containing 3.8% flakes and 2% CNT.

**Figure 4 materials-15-08201-f004:**
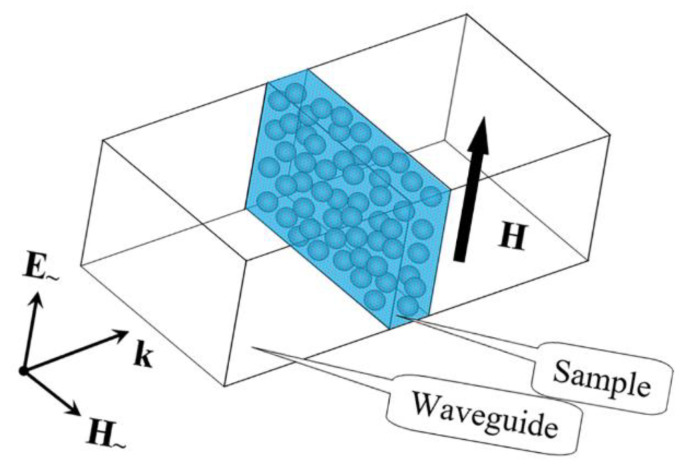
Scheme of microwave measurements.

**Figure 5 materials-15-08201-f005:**
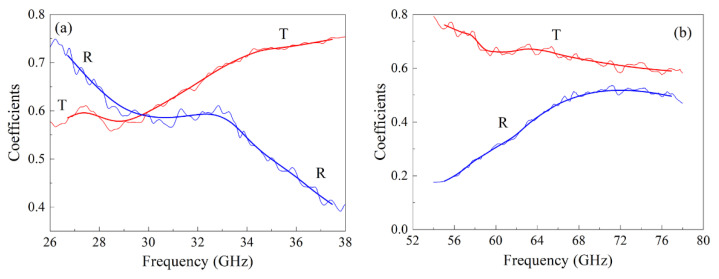
Frequency dependences of transmission and reflection coefficients for a composite with 3.8% flakes and 2% CNT in the frequency range from 26 to 38 GHz (**a**); in the frequency range from 53 to 78 GHz (**b**).

**Figure 6 materials-15-08201-f006:**
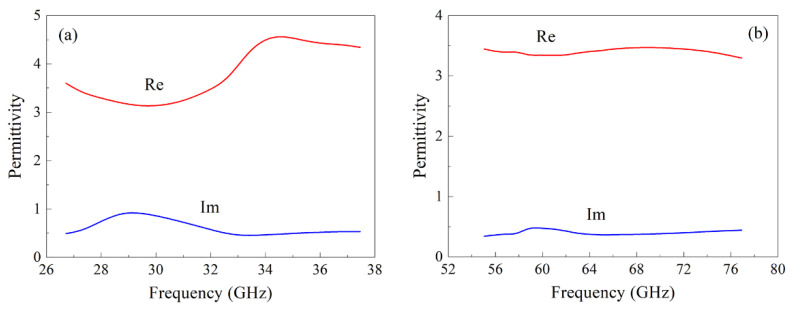
Frequency dependences of the permittivity of the composite with 3.8% flakes and 2% CNT in the frequency range from 26 to 38 GHz (**a**); in the frequency range from 53 to 78 GHz (**b**).

**Figure 7 materials-15-08201-f007:**
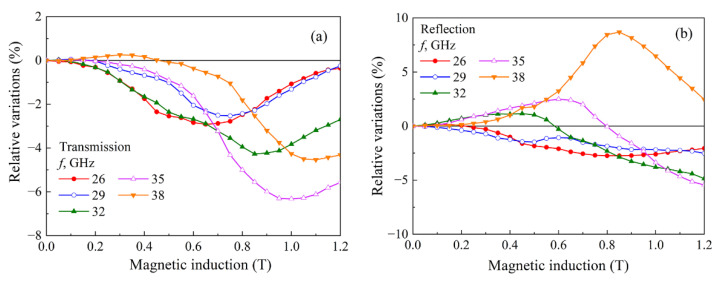
Dependences of the transmission coefficient (**a**) and reflection coefficient (**b**) on the magnetic field for a composite with 3.8% flakes and 2% CNT.

**Figure 8 materials-15-08201-f008:**
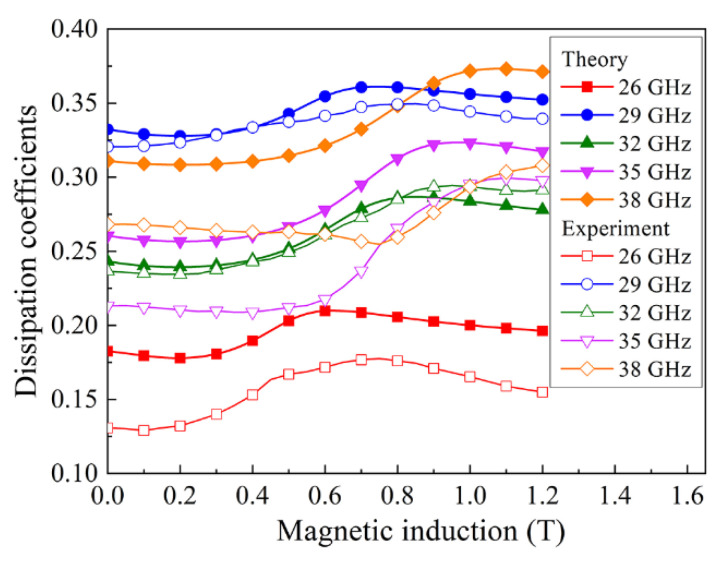
Dependences of microwave dissipation on the magnetic field for several millimeter-wave frequencies for a composite with 3.8% flakes and 2% CNT.

**Figure 9 materials-15-08201-f009:**
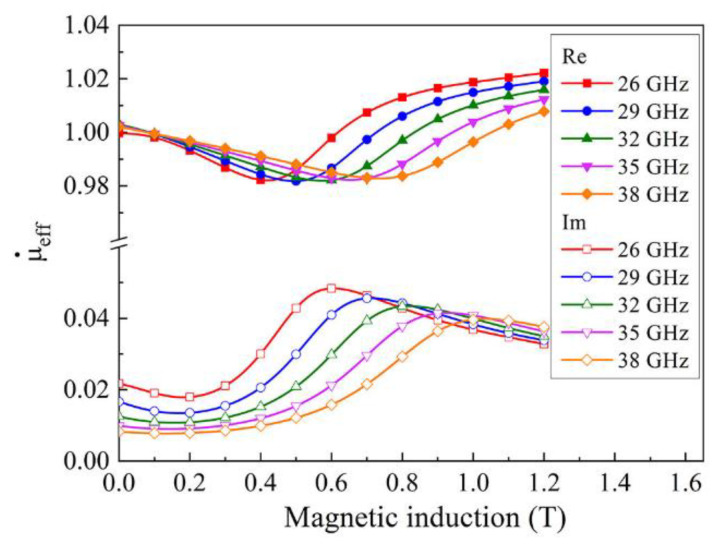
The dependence of the effective magnetic permeability on the magnetic field, calculated for several frequencies for a composite with 3.8% flakes and 2% CNT.

**Figure 10 materials-15-08201-f010:**
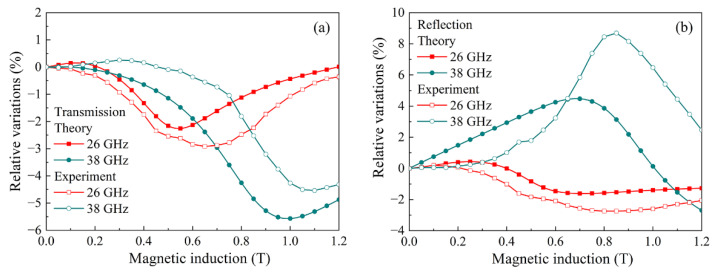
Comparison of theoretical and experimental dependences of the transmission coefficient (**a**) and reflection coefficient (**b**) on the magnetic field for a composite with 3.8% flakes and 2% CNT.

**Table 1 materials-15-08201-t001:** The content of elements in a Finemet alloy particle.

Element	Content (wt. %)
C	48.03
O	5.22
Al	0.72
Si	6.06
Nb	1.70
Cl	0.53
Ca	0.41
Fe	33.48
Ni	3.86
Matrix	Correction

**Table 2 materials-15-08201-t002:** Summary of estimates of the complex permittivity results and specific electrical conductivity of the samples, averaged over the measurement frequency ranges.

Frequency Range (GHz)	Sample Name	*ε*′	*ε*″	*σ* (S/m)
12–17	Epoxy matrix	3.2	0.23	–
	Composite 15%	7.5	3.13	2.45
17–26	Epoxy matrix	2.9	0.11	–
	Composite 15%	8.2	1.5	1.9
26–38	Epoxy matrix	2.6	0.33	–
	Composite 15%	5.4	1.1	2.0

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
