# Peer review of "The Microwave Absorption in Composites with Finemet Alloy Particles and Carbon Nanotubes"

_materials, 2022, doi:10.3390/ma15228201_

Round 1
Reviewer 1 Report
Comments on materials-2036299
The manuscript entitled “The Microwave Absorption in Composites with Finemet Alloy Particles and Carbon Nanotubes” studied the absorption of waves of the centimeter and millimeter wavebands in composites with Finemet alloy particles and carbon nanotubes. FMR and FMAR are observed in such composites. A method is proposed for calculating the effective dynamic magnetic permeability of a composite with different distributions.
The manuscript is well-written, however, there are several amendments required to be resolved before possibly accepting it for publication which are disclosed below:
· What do the authors mean by “groups of particles” in the abstract? It is suggested to discuss more for better understanding if a reader reads only an abstract.
· The introduction has some flaws, a more detailed novelty of their work should be clearly addressed. Thorough references should be cited in the Introduction, which is missing. Please provide relevant references for the statement “Landau-Lifshitz-Gilbert equation in conjunction with Bruggeman's method” (Line 35).
· The authors need to show what specific kind of Epoxy resin they have used to prepare the composite samples.
· Please check this sentence “the size of the cavity in which exactly equals to the transverse dimensions of the waveguides in which microwave measurements will be carried out”. It sounds not correct.
· How could authors just say that the curing of samples was done “within a few hours”, however, every polymer has its own curing temperature.
· Why do the authors specifically use these two series of composite samples with 15 wt. % flakes and 3.8 wt. % flakes + 2 % CNT ???
· The authors need to mention the composite name in the figure caption of figure 1. They can't just use the word “composite medium”.
· It is highly suggested to show the EDS elemental mapping to assure the content of elements in their alloy.
· How do authors assure that there is no electrical contact between the particles? Please clarify with some relevant references.
· The SEM images are of very poor resolution, it is recommended to provide high-resolution SEM images with different magnifications and proper labeling. Even the scale bar is not visible. Please check carefully. It is also suggested to provide SEM images of mere particles.
· How do the authors assure the dispersion of fillers in the polymer? The aggregation of particles will affect the final properties. Why didn't the authors further increase the filler concentration in the matrix? The authors should provide some experiments to prove it.
· Why do the authors show the MH loop of a composite material containing 3.8 % flakes and 2 % CNT?? Why not with and without polymer to understand the effect of polymer on the particles as well?
· What calibration technique the authors used to calibrate the VNA for the measurements?? Please write a sentence over it with a relevant reference.
· It is suggested to increase the thickness of the curves of the main results, it seems a little blurry to a reader.
· The authors should add the results of merely epoxy composite as well and compare it with filler composites.
· The authors should provide more discussions on the mechanisms for performance strategies, which would be beneficial for readers to understand their significance.
· The authors need to provide a comparison of their work with other reported ones to assure the novelty of their work and also propose some prospective applications of their work in the last paragraph before the Conclusions which are lacking in their submitted manuscript.
· It is also recommended to add some references from recent years of the related work which is missing in their reference list. Some references are important to understand the progress of polymer-based composite materials and their advantages: Composites Part A, 2022, 153 (2022) 106734; EPJ Appl. Metamat. 8, (2021) 10.
· The conclusions are not informative. They do not contain numerical data and must be carefully rewritten. Moreover, the authors are recommended to add a few sentences about the prospective applications of their proposed work.
· The manuscript contains a few misprints, and lost intervals and should be corrected in this respect.
Author Response
The authors are very thankful to the reviewer for his/her attention to our work and many useful comments and recommendations. We tried to take into account all of them and introduced many corrections. We regard all recommendations as fair and reasonable. In some cases, however, we can’t follow to the recommendations literally since it could excessively increase of volume of the paper. In such cases, we confine ourselves by short corrections and additional references, if this case is known from literature. We believe that now the paper looks better and more convincible.
The detailed reply to the reviewer’s comments can be find in the attached file.

Reviewer 2 Report
Please see the enclosed comments file.

Author Response

(The authors gave the same response as above.)

Reviewer 3 Report
Rinkevich et al. have presented the manuscript titled: The Microwave Absorption in Composites with Finemet Alloy Particles and Carbon Nanotubes. Overall presentation of the proposed work is good, but there requiring some modifications which I think are necessary to explain before publication.
1. The portion of abstract is weak, I suggest the authors to highlight their achievements (best result values) to attract the readers.
2. Please separate the section of fabrication and characterization. XRD and SEM analysis should be in the result and discussion section. Add the specifications of XRD, SEM or other eqipments in the characterization section.
3. Please provide the both XRD patterns of the samples with 15 wt. % flakes and 3.8 wt. % flakes and compares the results with the standard PDF Card# of previously reported data.
4. Why authors have only used 2 % CNT, have they tried with some other concentrations of CNT?
5. How authors have calculated the lattice parameters 2.871 Å and 2.841 Å, have they simulated the XRD patterns by some software? Has 2 % CNT affected the structure of the specimen? If not describe the reason.
6. In the line 120, authors state “structure of the composite with 15 % particles….” But in the caption of Figure 2 they describe “Structure of the composite medium with 3.8 %....” please describe because its confusing.
7. Figure 2b is not clear according to the standards of publication. I suggest the authors to replace it.
8. In Figure 5 and Figure 6, Is this the direct machine provided data from LCR or the best values at specific frequencies? I suggest the authors to add the data from the machine instead of few specific frequency points.
9. I suggest the authors to also provide the comparison study of 15 wt. % flakes and 3.8 wt. % flakes simultaneously. It will be better if authors can add one more sample between 3.8% and 15% flakes, which can be 7 %. In my opinion this study will become complete by doing such work.

Author Response

(The authors gave the same response as above.)

Round 2
Reviewer 1 Report
The authors have responded to all the comments. the reviewer has no further queries. The manuscript can be accepted for publication.